# Addition of Polyphenols to Drugs: The Potential of Controlling “Inflammaging” and Fibrosis in Human Senescent Lung Fibroblasts In Vitro

**DOI:** 10.3390/ijms25137163

**Published:** 2024-06-28

**Authors:** Maria Carolina Ximenes de Godoy, Gabriela Arruda Monteiro, Bárbara Hakim de Moraes, Juliana Alves Macedo, Gisele Mara Silva Gonçalves, Alessandra Gambero

**Affiliations:** 1School for Life Sciences, Pontifical Catholic University of Campinas (PUC-Campinas), Av. John Boyd Dunlop, Campinas 13034-685, SP, Brazil; maria.cxg@puccampinas.edu.br (M.C.X.d.G.); gabriela.am8@puccampinas.edu.br (G.A.M.); barbara.hm1@puccampinas.edu.br (B.H.d.M.); gmsg@puc-campinas.edu.br (G.M.S.G.); 2Department of Food and Nutrition, School of Food Engineering, State University of Campinas, Campinas 13083-862, SP, Brazil; jumacedo@unicamp.br

**Keywords:** senotherapeutic, senomorphic, senostatic, senescence-associated secretory phenotype, fibrosis, chronic lung diseases, aging

## Abstract

The combination of a polyphenol, quercetin, with dasatinib initiated clinical trials to evaluate the safety and efficacy of senolytics in idiopathic pulmonary fibrosis, a lung disease associated with the presence of senescent cells. Another approach to senotherapeutics consists of controlling inflammation related to cellular senescence or “inflammaging”, which participates, among other processes, in establishing pulmonary fibrosis. We evaluate whether polyphenols such as caffeic acid, chlorogenic acid, epicatechin, gallic acid, quercetin, or resveratrol combined with different senotherapeutics such as metformin or rapamycin, and antifibrotic drugs such as nintedanib or pirfenidone, could present beneficial actions in an in vitro model of senescent MRC-5 lung fibroblasts. A senescent-associated secretory phenotype (SASP) was evaluated by the measurement of interleukin (IL)-6, IL-8, and IL-1β. The senescent-associated β-galactosidase (SA-β-gal) activity and cellular proliferation were assessed. Fibrosis was evaluated using a Picrosirius red assay and the gene expression of fibrosis-related genes. Epithelial-mesenchymal transition (EMT) was assayed in the A549 cell line exposed to Transforming Growth Factor (TGF)-β in vitro. The combination that demonstrated the best results was metformin and caffeic acid, by inhibiting IL-6 and IL-8 in senescent MRC-5 cells. Metformin and caffeic acid also restore cellular proliferation and reduce SA-β-gal activity during senescence induction. The collagen production by senescent MRC-5 cells was inhibited by epicatechin alone or combined with drugs. Epicatechin and nintedanib were able to control EMT in A549 cells. In conclusion, caffeic acid and epicatechin can potentially increase the effectiveness of senotherapeutic drugs in controlling lung diseases whose pathophysiological component is the presence of senescent cells and fibrosis.

## 1. Introduction

Recent studies have demonstrated the accumulation of senescent cells in the lung in chronic lung diseases, such as chronic obstructive pulmonary disease (COPD) and idiopathic pulmonary fibrosis (IPF) [1]. Cellular senescence is a cellular fate characterized by irreversible cell cycle arrest associated with the development of a senescence-associated secretory phenotype (SASP) that drives chronic inflammation (inflammaging) and fibrosis, and, in addition, converts neighboring cells to the senescence state and is linked to the promotion of lung cancer [2,3,4]. Identifying senescent cells in chronic lung diseases associated with a possible role of SASP cytokines and fibrotic inducers in their physiopathology have increased the enthusiasm about a possible therapeutic alternative based on senotherapeutic drugs [2]. Potential therapeutic alternatives include senolytics drugs that act to eliminate senescent cells or senostatic or senomorphic drugs that block or modulate the aging phenotype. Lung and other “senopathies” [5] are undergoing clinical trials to verify the safety and efficacy of senolytic drugs. Participants with IPF who received the combination of dasatinib and quercetin for three weeks or a placebo experienced no adverse effects. However, the frailty, lung, or physical function of patients did not differ significantly between the groups [6]. A phase I clinical trial with older Alzheimer patients is also ongoing, revealing the safety of dasatinib and quercetin and the presence of dasatinib in cerebrospinal fluid (CSF). The cognitive and neuroimaging outcomes did not differ significantly post-treatment, but the CSF levels of interleukin-6 (IL-6) and glial fibrillary acidic protein (GFAP) increased, although several other SASP-associated cytokines decreased [7]. Clinical trials with another senolytic drug, fisetin, are ongoing (www.clinicaltrials.org).

Drugs with senomorphic activity include natural compounds such as apigenin, epigallocatechin-3-gallate, avenanthramide, resveratrol, and old synthetic drugs such as metformin and rapamycin (sirolimus) [8]. However, the main challenge of senotherapeutics development is the heterogeneity of senescence phenotypes. Metformin was able to limit the lung damage induced by radiotherapy in a preclinical mouse model and in lung epithelial cells in vitro by decreasing *Ccl2* expression, targeting AKT-p53/p21 signaling, and reducing reactive oxygen species (ROS) accumulation, respectively [9]. Lung epithelial cell senescence was elevated after lipopolysaccharide (LPS) exposure in vivo and in vitro, and metformin was able to protect epithelial cells by increasing the expression of ATG5 and the autophagy activity [10]. Rapamycin is described by its efficacy in suppressing senescence and the SASP due to the activation of autophagy [11]. The inhibition of epithelial cell senescence by rapamycin suppressed the activation of pulmonary fibroblasts and attenuated pulmonary fibrosis [12]. While senolytics are already in phase 1 clinical studies, progress in understanding the potential of senomorphic drugs has been based on observations of the possible antiaging potential of metformin, for example [13].

Polyphenols are natural chemical compounds characterized by multiple phenol units. They have antioxidant and anti-inflammatory activities associated with prolonging lifespan by inhibiting chronic inflammation signaling, influencing gene expression, and causing epigenetic modifications [14]. Quercetin and caffeic acid increased the Sirt expression in old fibroblast cells, suggesting that it could be the antiaging mechanism [15]. Polyphenols are also described as potential antifibrotic compounds, such as quercetin, hesperitin, resveratrol, and epigallocatechin-3-gallate, being able to improve cardiac fibrosis [16]. Quercetin reduced extracellular matrix (ECM) deposition was conducted by inhibiting TGF-β signaling in a model of heart dysfunction in vivo [17]. Quercetin also inhibited nuclear factor-kappaB (NF-κB) and stimulated nuclear factor erythroid 2-related factor 2 (Nrf-2) signaling in the heart tissue of obese rats, preventing cardiac remodeling [18]. Resveratrol also demonstrated potential in a lung fibrosis experimental model by reducing inflammatory and fibrotic signaling pathways such as Toll-like receptor 4 (TLR4)/NF-κB and TGF-β1/smad3, but also by modulating epithelial–mesenchymal transition (EMT) [19,20,21].

Considering that the therapeutic options for the management of lung fibrosis are restricted to the two antifibrotic drugs, pirfenidone and nintedanib, which have a high cost, the combination of polyphenol and a drug is pointed to as a senolytic option at the moment; therefore, we searched for combinations of old drugs (including pirfenidone and nintedanib) and polyphenols for antifibrotic and senomorphic activity using human senescent lung fibroblast in vitro because lung fibroblast determines the severity and outcomes of several chronic lung diseases [22]. Cellular senescence was induced by doxorubicin in human lung fibroblast MRC-5 cells. Doxorubicin is a topoisomerase II inhibitor used therapeutically as an anticancer drug. Doxorubicin induces DNA damage, senescence-associated β-galactosidase (SA-β gal), SASP, and cell-cycle arrest in several cell lines, demonstrating an ability to induce senescence [23,24]. In addition, a brief assay of EMT was performed using human lung epithelial A549 cells and TGF-β, considering that EMT is a key driver of the process of pulmonary fibrosis [25].

## 2. Results

### 2.1. Senomorphic Effects of Drugs and Polyphenols

The main cytokine associated with senescence, IL-6, was inhibited by metformin and some polyphenols such as caffeic acid, epicatechin, gallic acid, and resveratrol in MRC-5 cells with doxorubicin-induced senescence (Figure 1A). The IL-8 production was inhibited by caffeic acid, chlorogenic acid, gallic acid, and assayed resveratrol polyphenols, while IL-1β was only inhibited by caffeic acid in senescent MRC-5 cells (Figure 1B,C).

When drugs and polyphenols were combined, metformin with caffeic acid inhibited IL-6 production by senescent MRC-5 cells (Figure 2A). Metformin with caffeic acid, chlorogenic acid epicatechin, gallic acid, and quercetin inhibited IL-8 production by senescent MRC-5 cells (Figure 2C), while no IL-1β inhibition was observed by metformin in combination with polyphenols (Figure 2E). IL-8 was inhibited by rapamycin with caffeic acid, chlorogenic acid, epicatechin, and quercetin used in the treatment (Figure 2D). In contrast, no IL-6 (Figure 2B) or IL-1β (Figure 2F) inhibition was observed by rapamycin or polyphenols.

Considering the ability of caffeic acid to inhibit the three SASP markers (IL-6, IL-8, and IL-1β) employed in this work and an additive response observed by caffeic acid and metformin in IL-6 inhibition, we investigate if other senescence markers, such as cell-cycle arrest and SA-β-gal activity, could be modified by it. In this experimental set, caffeic acid, metformin, or a combination of both were co-incubated with doxorubicin, and the proliferation rate was evaluated after 24 h, 6 days, or 20 days. As we demonstrate in Figure 3A, senescent MRC-5 cells had a high proliferation rate after 6 days, but not after 24 h or 20 days of doxorubicin exposure, when it was co-incubated with metformin, but not caffeic acid alone or a combination of metformin and caffeic acid. The cyclin-dependent kinase (CDKN) inhibitors’, p16 and p21, gene expressions were evaluated at this experimental point, and we could observe a significative inhibition of gene CDKN2A (p16) in senescent MRC-5 cells by caffeic acid, metformin, or both (Figure 3B). No CDKN1A (p21) gene expression inhibition was registered (Figure 3C). MRC-5 senescent cells express more CDKN2A (0.089 ± 0.005 and 0.340 ± 0.023 AU for control and senescent cells, respectively; *p* < 0.05) and CDKN1A (13.17 ± 0.6 and 22.30 ± 0.34 AU for control and senescent cells, respectively; *p* < 0.05).

The senescence-associated β-galactosidase (SA-β-gal) activity was also evaluated after 6 and 20 days of doxorubicin co-incubation with metformin, caffeic acid, or a combination of both in senescent MRC-5 cells. The activity of SA-β-gal was reduced on the sixth day in all experimental treatments (Figure 4). Control cells did not show SA-β-gal activity.

### 2.2. Evaluation of the Antifibrotic Activity of Drugs and Polyphenols

Senescent MRC-5 cells produced more collagen fibers that were stained by Picrosirius red than control MRC-5 (0.174 ± 0.02 and 0.396 ± 0.063 Ratio OD Picrosirius red and Crystal violet for control and senescent MRC-5 cells, respectively; *p* < 0.05). An antifibrotic activity was observed using Picrosirius red evaluation in senescent MRC-5 cells only by isolated epicatechin (Figure 5).

The antifibrotic effect was observed when epicatechin was combined with metformin, nintedanib, pirfenidone, and rapamycin (Figure 6A–D). As epicatechin was the most effective antifibrotic treatment, the expression of COL3A, COL1A, and TGFB1 was evaluated in senescent MRC-5 cells. As demonstrated in Figure 7A, epicatechin alone or combined with metformin, pirfenidone, and rapamycin inhibits the expression of COL3A in senescent MRC-5 cells. Senescent cells express COL3A1 similarly to control cells (0.209 ± 0.009 and 0.323 ± 0.068 AU for control and senescent MRC-5 cells, respectively; *p* = 0.5221). Control cells express COL1A at a very low level; in senescent cells, the expression was undetectable (0.0000037 ± 0.000004 UA for control cells). No differences were observed in COL1A gene expression when senescent cells were treated with epicatechin or drugs, but the expression of COL1A could be detected after the treatment (Figure 7B). The TGFB1 expression was also evaluated, and epicatechin alone or combined with metformin, pirfenidone, and rapamycin inhibited its expression in senescent MRC-5 cells (Figure 7C). The expression of the TGFB1 was significantly higher in senescent MRC-5 cells when compared to control cells (0.013 ± 0.001 and 0.225 ± 0.087 AU for control and senescent MRC-5 cells, respectively; *p* < 0.01).

### 2.3. EMT Evaluation

A549 epithelial lung cells were incubated with TGF-β1 for 48 h, and a reduction in the CDH1 gene was observed (26.7 ± 2.9 and 1.0 ± 0.1 AU for control or TGF-β1-treated A549 cells, respectively; *p* < 0.01). No significant alteration was observed in the ACTA1 gene, responsible for producing α-smooth muscle actin (SMA) protein as a fibroblast marker (1.7 ± 0.1 and 1.0 ± 0.2 for control or TGF-β1-treated A549 cells, respectively). Epicatechin and nintedanib increased the epithelial marker CDH1 (Figure 8A), while no significant alterations were observed by epicatechin or drugs in ACTA1 gene expression (Figure 8B).

## 3. Discussion

Cellular senescence is not restricted to chronological aging but can be induced at any age by stress responses induced by acute or chronic, extrinsic, and intrinsic stimuli. Cellular senescence is essential in developing chronic inflammatory diseases, cancer, and age-related diseases. This detrimental low-grade inflammation called “inflammaging,” resulting from a decline in immune function, favors the aging of other cells and modifies the microenvironment by increasing pro-oxidant and fibrosis responses [26], as observed in several lung diseases. Therefore, strategies that impact inflammation could help change the accumulation or the activity of senescent cells and improve or treat chronic lung diseases, as is demonstrated by senolytic and senomorphic drugs. Senomorphic activity is described in drugs that present, as their mechanism of action, a potent radical scavenger or anti-inflammatory activity, as found in natural polyphenols or drugs already used therapeutically such as metformin [26].

Using a cell line of human lung fibroblasts, we induced cellular senescence and searched for combinations of drugs and natural polyphenols to control inflammaging and fibrosis responses in this model. A combination of a drug, dasatinib, and a natural polyphenol, quercetin, has proven to be the best option for senolytics and is in initial clinical studies [6,27]. The most promising combination we tested for controlling inflammation associated with senescent cells was metformin and caffeic acid, which inhibits IL-6 and IL-8 and maintains IL-1β levels. IL-6 is the most prominent cytokine of SASP [28,29]. The fact that metformin demonstrated a senomorphic effect corroborates previous studies [29,30] that associated the antiaging effect with downstream consequences of AMP-activated protein kinase (AMPK) activation, inhibition of the mammalian target of rapamycin complex 1 (mTORC1), and inhibition of the production of mitochondrial ROS, but which also noted its poorly understood immunomodulatory activity (for review see [31]). Metformin, at doses comparable to our work, could inhibit IL-6 gene expression in fibroblasts expressing oncogenic ras by inhibiting NF-κB activation [31]. Caffeic acid also demonstrates excellent results in inhibiting SASP in senescent MRC-5 cells. Caffeic acid, a hydroxycinnamic acid, is widely distributed in fruits and vegetables whose extracts show anti-inflammatory activity [32,33,34]. In the senescence context, only the caffeic acid derivative methyl ester, but not caffeic acid, demonstrated the ability to inhibit IL-6 and IL-8 release in BJ fibroblasts treated with bleomycin by a NF-κB-inhibition mechanism [35]. Of note, we did not observe a synergistic effect between metformin and caffeic acid in our model; it was only an additive effect that encouraged us to use the combination in future studies. The inhibitory effect of SASP by methyl caffeate reported by Lim and collaborators was not associated with alterations of other senescence markers such as p21 expression [35]. In our results, only metformin, not caffeic acid, could reverse the increased expression of p16 and restore cellular proliferation when co-incubated with doxorubicin during senescence induction in MRC-5 cells. However, the combination reduced SA-β-gal activity, suggesting that it could be protective in senescence induction. The SA-β-gal activity was lower in diabetic kidney disease-mesenchymal stem cells from participants with diabetes mellitus who were on metformin therapy, demonstrating that it is possible to revert or delay the induction of senescence [36]. Experimentally, in a preclinical mouse model of radiation-induced pneumopathy, metformin was able to limit Cdkn1a/p21 expression, reduce SA-β-gal activity in epithelial cells, and reduce the expression of SASP markers, as well as decreasing expression levels, mainly Ccl2 and Cxcl1. However, the IL6 gene expression was not significantly reduced, and Mmp2 was increased [9]. Some polyphenols promoted increased IL-1β levels in senescent fibroblast cultures. These results must be better explored in the context of a normal cell’s senescence. Our initial hypothesis is that it may be related to the apoptotic or autophagic mechanism described for some polyphenols [37,38].

The control of SASP is interesting in the context of lung diseases because it has been suggested that it plays a central role in initiating and progressing pneumonitis and pulmonary fibrosis. Thus, using senescent lung fibroblasts in vitro, we evaluated the potential of isolated polyphenols and combined them with senomorphic or antifibrotic drugs to control collagen production as a fibrosis model. Only epicatechin isolated and in combination with metformin, nintedanib, pirfenidone, and rapamycin demonstrated the potential to prevent the fibrosis process in senescent fibroblasts. Epicatechin isolated and in combination with metformin, pirfenidone, and rapamycin reduces the expression of COL3A and TGFB1 in our model. Some plant extracts, such as *Rosa sterile* [39] and *Uncaria gambir* [40], abundant in catechins, demonstrated antifibrotic effects in experimental models of bleomycin-induced lung fibrosis [41]. These antifibrotic effects are shared with other catechins, such as epicatechin gallate (ECG) or epigallocatechin gallate (EGCG); one advantage is that epicatechin is more bioavailable than ECG or EGCG when orally administered [42,43]. The mechanism involved in the antifibrotic effects of catechins is related to their antioxidant and anti-inflammatory properties and their ability to suppress the activation of AKT/mTOR and Smad2/3 signaling pathway in lung tissues, which would inhibit the phenomenon of EMT [39,44]. We confirm that epicatechin restored the gene expression of the CDH1 gene, responsible for producing epithelial marker E-cadherin, in the A549 cell line exposed to TGF-β as a model of EMT. The same effect was observed with nintedanib. Although nintedanib does not inhibit the gene expression of TGFB1 in senescent fibroblasts, the ability to inhibit EMT in A549 cells exposed to TGF-β is described for this drug [45].

It is common to think about the structure–activity relationship concerning polyphenols. Although they are plant-specialized (secondary) metabolites containing an aromatic (benzene or phenol) ring with one or more hydroxyl groups in their molecule, their structural diversity is broad, and not all molecules share the same biological activities [46]. Our study presented data obtained from in vitro approaches that allowed testing various combinations of senescent fibroblasts and the EMT process. We cannot fail to consider that other polyphenols could also have presented senomorphic activity, as we investigated only one of them in more detail, caffeic acid. And considering the approach to fibrotic diseases associated with aging, senomorphic and antifibrotic actions are not concentrated in a single polyphenol, having a hydroxycinnamic acid and a flavonol demonstrating the potential to be exploited in an association. Another challenge in searching for senolytics and senostatics is the great diversity of senescent phenotypes and drugs, which only sometimes maintain efficacy in different cell players in one disease. Given this diversity, in vivo studies of lung aging models for proof of concept are necessary. On the other hand, as these are widely known drugs and polyphenols with an already established safety profile, phase 1 clinical studies can be considered to obtain this proof of concept.

In conclusion, in the context of lung cellular senescence, our experiments indicate that caffeic acid and epicatechin may have the potential to be co-administered with drugs already in use in current therapy to control the production of SASP mediators, and the establishment and progression of senescence and fibrotic responses associated with senescence.

## 4. Materials and Methods

### 4.1. MRC-5 Cell Culture

The MRC-5 (human lung fibroblast cells) was purchased from the Rio de Janeiro Cell Bank (BCRJ 0180; OS.C.5930.21; Lot:0001246; Passage:21). Cells were cultured in DMEM supplemented with 1% non-essential amino acids, 10% fetal bovine serum (FBS), 1% penicillin, streptomycin, and amphotericin solution in a humidified incubator at 37 °C under an atmosphere of 5% CO_2_. All culture reagents were purchased from Invitrogen (Carlsbad, CA, USA).

### 4.2. MRC-5 Senescence Induced by Doxorubicin

MRC-5 cells were incubated with doxorubicin 1.5 µM (Merck, Saint Louis, MO, USA) for 24 h, as described before [35]. Cells were cultivated as described above without adding doxorubicin, and experiments were performed after 6 or 20 days of doxorubicin incubation. Controls were performed with MRC-5 passage 22 to avoid replicative senescence. Doxorubicin-induced senescence in MRC-5 cells was characterized by a high expression of SA-Gal, cell-cycle arrest, and high IL-6 production after 20 days of cell culture [46].

### 4.3. Determination of Drug and Polyphenol Non-Toxic Concentrations

The non-toxic concentration was determined in control cells by using pirfenidone (0.1; 0.4; 0.8; 1.6 mM; Merck), nintedanib (0.1; 0.5; 1; 10 µM; Merck), metformin (0.5; 1; 5; 10; 50 mM; NC Pharma, Hortolândia, Brazil), and rapamycin (2, 10, 20, 100, 200 nM; Merck) incubation for 24 h followed by an MTT assay. The dose ranges were chosen after a brief literature review about effective or toxic doses of drugs in different in vitro cell cultures. For non-toxic concentration determination, the culture medium was removed, and the MTT solution (0.5 mg·mL^−1^ in PBS) was added for two hours at 37 °C under an atmosphere of 5% CO_2_. The formazan crystals were solubilized in isopropanol for 10 min, and absorbance was read at 540 nm (Berthold technology; Bad Wildbad, Germany). Pirfenidone, nintedanib, and rapamycin were dissolved in dimethyl sulfoxide (DMSO), followed by dilutions in DMEM. The DMSO final concentration in cell experiments was less than 0.1% (toxicity of DMSO in MRC-5 was observed for higher concentration than 0.5%). Metformin was dissolved in DMEM. Each experiment was performed in triplicate with one repetition. The cell viability was also evaluated in control cells incubated for 24 h with different concentration ranges of the polyphenols caffeic acid (10; 50; 100; 500; 1000 µM), chlorogenic acid (10; 50; 100; 500; 1000 µM), epicatechin (10; 50; 100; 500; 1000 µM), gallic acid (10; 50; 100; 500; 100 µM), quercetin (5; 10; 50; 100 µM), and resveratrol (1–1000 µM). Chlorogenic acid and gallic acid were dissolved in DMEM, and all the others were dissolved in DMSO, as described before. All polyphenols used were purchased from Merck. The highest non-toxic concentration for each drug or polyphenol was combined and cytotoxicity was assessed in control and senescent cells. Only non-toxic concentrations were used in the experiments. For non-toxic concentration data see Appendix A and Supplementary Materials from de Godoy et al. [46].

### 4.4. SASP Evaluation

The release of IL-6, IL-8, and IL-1β was quantified in the supernatant of senescent cells after 24 h incubation in DMEM without FBS with drugs or polyphenols isolated by using EIA kits from BD (BD OptEIA human IL-6, IL-8, and IL-1β kit, San Diego, CA, USA). At the end of the experiment, an MTT assay was performed to evaluate cell viability, as described in Section 4.3.

### 4.5. Senescence-Associated β-Galactosidase (SA-β-gal) Assay

For SA-β-gal activity, cells were fixed with 3.7% formaldehyde in phosphate-buffered saline (PBS) for 3 min. After washing with PBS, cells were stained with a solution containing 1 mg/mL of X-gal (Invitrogen), 40 mM citric acid/sodium phosphate buffer (pH 6.0), 5 mM potassium ferricyanide (Merck), 5 mM potassium ferrocyanide (Merck), 150 mM NaCl, and 2 mM MgCl_2_ for six hours at 37 °C [47]. Cells were washed with PBS and photographed in five random fields for cell count.

### 4.6. Quantitative Analysis of Gene Expression

For CDKN2A and CDKN1A gene expression analysis, mRNA was purified using an RNeasy Plus kit (Qiagen, Hilden, Germany). Real-time PCR was performed in a QuantStudio 1 real-time PCR system (Applied Biosystems, Waltham, MA, USA). cDNA was synthesized using the High Capacity cDNA Reverse Transcription kit (Thermo Fisher, Waltham, MA, USA). Quantitative RT-PCR was performed using SybrGreen Master Mix (Thermo Fisher) and primers described in Appendix A. The expression of b-actin rRNA was used as an endogenous control for data normalization. The results were analyzed using the 2^−ΔΔCt^ relative quantification method. 

### 4.7. Picrosirius Red and Crystal Violet Stains

For collagen evaluation, cells were fixed with Bouin’s solution for 1 h at room temperature. After removal of Bouin’s solution, cells were incubated with Picrosirius Red dye solution (1 mg·mL^−1^ Picrosirius Red in water-saturated picric acid) for 1 h at room temperature. Cells were washed four times with 0.01 M HCl, and the dye was eluted with 0.1 M NaOH by mixing on an orbital shaker for 30 min. The eluted dye was measured at the absorbance of 550 nm with a plate reader. Cells were stained with the DNA-binding dye Crystal Violet to assess differences in cell density after Picrosirius red elution. Crystal violet solution (0.1% Crystal violet in water) was added to each well and incubated for 30 min at room temperature. After washing, Cristal violet was eluted with methanol. The eluted dye was measured at the absorbance of 540 nm with a plate reader [48]. COL3A, COL1A, and TGF-β gene expression analysis were performed as described in Section 4.6.

### 4.8. Epithelial-Mesenchymal Transition (EMT) Assay Using A549 Cells

The A549 cells (human lung carcinoma epithelial cells) were purchased from the Rio de Janeiro Cell Bank (BCRJ 0033; OS.C.6429.22; Lot:001313; Passage:88) and cultured in DMEM supplemented with 10% FBS and 1% penicillin, streptomycin, and amphotericin solution. Cells were incubated for 72 h with TGF-β (10 ng/mL; Peprotech-Thermofisher, Cranbury, NJ, USA) in serum-free culture media. The EMT was evaluated by gene expression of CDH1 and ACTA1 [49]. Epicatechin, metformin, nintedanib, pirfenidone, and rapamycin doses were tested for cytotoxicity before MTT assay experiments. Only non-toxic doses were employed. Data on cytotoxicity are presented in the Appendix A.

### 4.9. Statistical Analysis

All data are expressed as mean ± S.E.M. Data comparisons were performed using a one-way analysis of variance followed by Tukey’s multiple comparison test. An associated probability (*p* value) of less than 0.05 was considered significant.

## Figures and Tables

**Figure 1 ijms-25-07163-f001:**
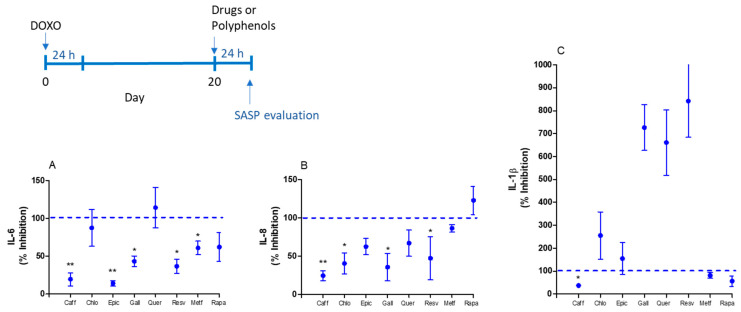
SASP evaluation by IL-6 (**A**), IL-8 (**B**), and IL-1β (**C**) basal production by senescent MRC-5 cells after 24 h of incubation with caffeic acid (Caff; 1000 µM), chlorogenic acid (Chlo; 1000 µM), epicatechin (Epic; 1000 µM), gallic acid (Gall; 100 µM), quercetin (Quer; 100 µM), resveratrol (Resv; 100 µM), metformin (Metf; 10 mM) and rapamycin (Rapa; 100 nM). Data are the mean and SEM of two experiments in triplicate. Data were normalized by control values (non-treated senescent cells) represented by the dashed line. * *p* < 0.05 and ** *p* < 0.01 compared to non-treated senescent cells.

**Figure 2 ijms-25-07163-f002:**
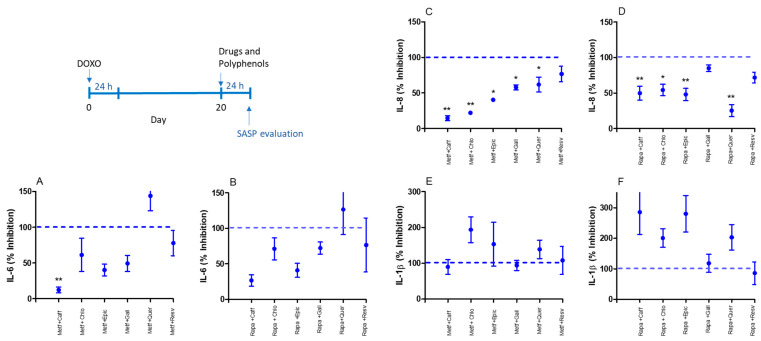
SASP evaluation by IL-6 (**A**,**B**), IL-8 (**C**,**D**), and IL-1β (**E**,**F**) basal production by senescent MRC-5 cells after 24 h of incubation with combinations of metformin (Metf; 10 mM) or rapamycin (Rapa; 100 nM) with caffeic acid (Caff; 1000 µM), chlorogenic acid (1000 µM), epicatechin (Epic; 1000 µM), gallic acid (Gall; 100 µM), quercetin (Quer; 100 µM), or resveratrol (Resv; 100 µM). Data are the mean and SEM of two experiments in triplicate. Data were normalized by control values (non-treated senescent cells) represented by the dashed line. * *p* < 0.05 and ** *p* < 0.01 compared to non-treated senescent cells.

**Figure 3 ijms-25-07163-f003:**
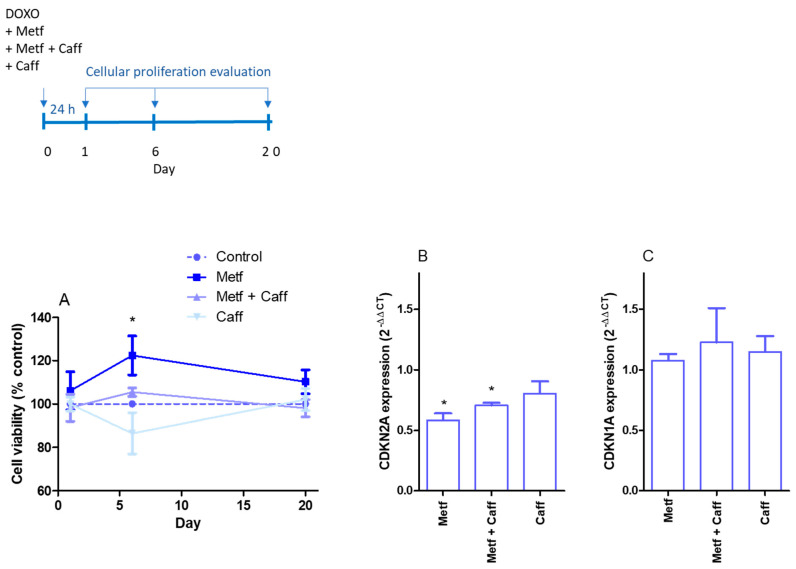
Proliferation of MRC-5 cells (**A**) after 1, 6, and 20 days from co-incubation of doxorubicin with metformin (Metf; 10 mM), metformin and caffeic acid (Metf 10 mM + Caff 1000 µM), or caffeic acid (Caff; 1000 µM). Gene expression of cyclin-dependent kinase (CDKN)2A (**B**) and CDKN1A (**C**) in MRC-5 cells after six days from co-incubation of doxorubicin with metformin (Metf; 10 mM), metformin and caffeic acid (Metf 10 mM + Caff 1000 µM), or caffeic acid (Caff; 1000 µM). (**A**) Data are the mean and SEM of two experiments in triplicate. Data were normalized by control values (non-treated senescent cells) represented by the dashed line. (**B**,**C**) Data are the mean and SEM of two experiments in duplicate normalized by the control group (non-treated senescent cells). * *p* < 0.05 when compared with non-treated senescent cells.

**Figure 4 ijms-25-07163-f004:**
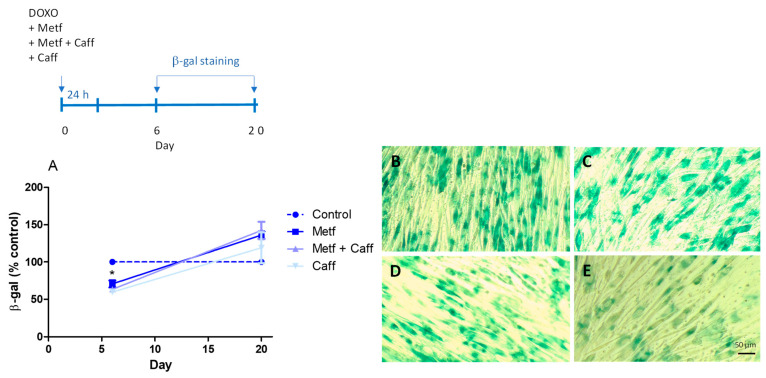
Senescence-associated β-galactosidase (SA-β-gal) staining of MRC-5 cells (**A**) after 6 and 20 days from co-incubation of doxorubicin with metformin (Metf; 10 mM), metformin and caffeic acid (Metf 10 mM + Caff 1000 µM), or caffeic acid (Caff; 1000 µM). Representative image of MRC-5 cells after six days from co-incubation of doxorubicin (**B**), doxorubicin with metformin (Metf, **C**), doxorubicin with metformin and caffeic acid (Metf + Caff, **D**), or doxorubicin with caffeic acid (Caff, **E**). (**A**) Data are the mean and SEM of two experiments in triplicate. Data were normalized by control values (non-treated senescent cells) represented by the dashed line. * *p* < 0.05 when compared with non-treated senescent cells.

**Figure 5 ijms-25-07163-f005:**
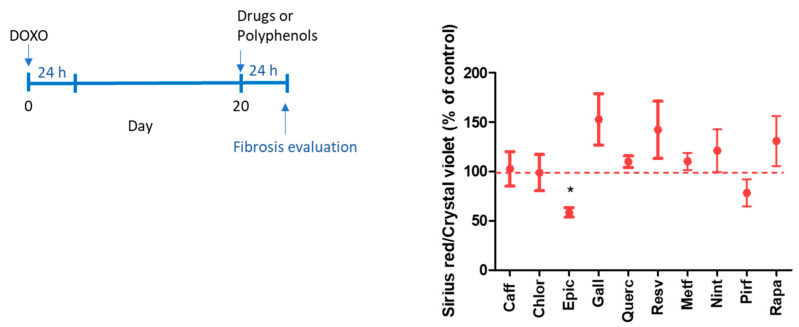
Fibrosis evaluation by collagen basal production by senescent MRC-5 cells after 24 h of incubation with caffeic acid (Caff; 1000 µM), chlorogenic acid (Chlo; 1000 µM), epicatechin (Epic; 1000 µM), gallic acid (Gall; 100 µM), quercetin (Quer; 100 µM), resveratrol (Resv; 100 µM), metformin (Metf; 10 mM), nintedanib (Nint; 1 µM), pirfenidone (Pirf; 0.8 mM), and rapamycin (Rapa 100 nM). Data are the mean and SEM of two experiments in triplicate. Data were normalized by control values (non-treated senescent cells) represented by the dashed line. * *p* < 0.05 when compared with control cells.

**Figure 6 ijms-25-07163-f006:**
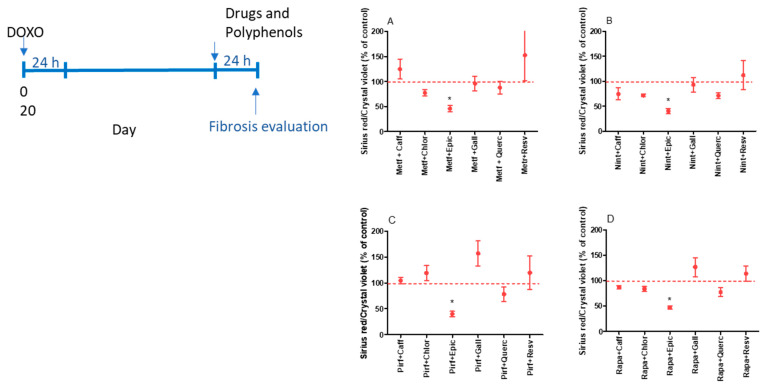
Fibrosis evaluation by collagen basal production by senescent MRC-5 cells after 24 h of incubation with combinations of metformin (Metf; 10 mM, **A**), nintedanib (Nint; 1 µM, **B**), pirfenidone (Pirf; 0.8 mM, **C**), and rapamycin (Rapa; 100 nM, **D**) with caffeic acid (Caff; 1000 µM), chlorogenic acid (Chlor; 1000 µM), epicatechin (Epic; 1000 µM), gallic acid (Gall; 100 µM), quercetin (Quer; 100 µM), or resveratrol (Resv; 100 µM). Data are the mean and SEM of two experiments in triplicate. Data were normalized by control values (non-treated senescent cells) represented by the dashed line. * *p* < 0.05 when compared with control cells.

**Figure 7 ijms-25-07163-f007:**
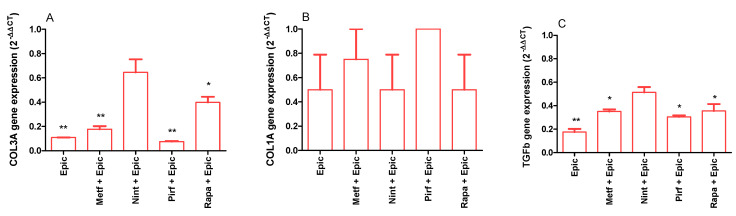
Gene expression of COL3A (**A**), COL1A (**B**), and TGFb (**C**) by senescent MRC-5 cells after 24 h of incubation with combinations of epicatechin (Epic; 1000 µM) or metformin (Metf; 10 mM), nintedanib (Nint; 1 µM), pirfenidone (Pirf; 0.8 mM), and rapamycin (Rapa; 100 nM) with epicatechin (Epic; 1000 µM). Data are the mean and SEM of two experiments in duplicate. Data were normalized by control values (non-treated senescent cells). * *p* < 0.05 and ** *p* < 0.01 when compared with control cells.

**Figure 8 ijms-25-07163-f008:**
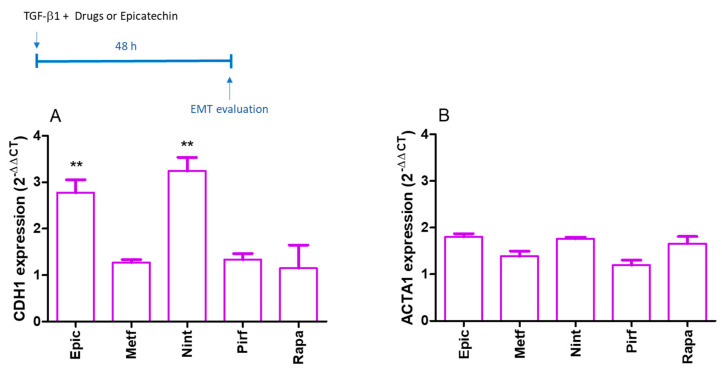
Gene expression of CDH1 (**A**) and ACTA1 (**B**) by A549 cells after 48 h of incubation with TGF-β1 (10 ng/mL) in the presence of epicatechin (Epic; 100 µM), metformin (Metf; 1 mM), nintedanib (Nint; 1 µM), pirfenidone (Pirf; 0.8 mM), or rapamycin (Rapa; 10 nM). data are the Mean and SEM of two experiments in duplicate. Control values are normalized data (A549 exposed to TGF-β1 only). ** *p* < 0.01 when compared with control.

## Data Availability

Data is contained within the article and Appendix A.

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
