# Peer review of "Addition of Polyphenols to Drugs: The Potential of Controlling “Inflammaging” and Fibrosis in Human Senescent Lung Fibroblasts In Vitro"

_ijms, 2024, doi:10.3390/ijms25137163_

Round 1

Reviewer 1 Report

Comments and Suggestions for Authors

Considering that pirfenidone and nintedanib are antifibrotic drug, and polyphenols are antioxidant and immunomodulatory natural products, this study explores the possible beneficial effect of combinations of those antifibrotic drugs and several polyphenols for preventing cellular senescence of human lung fibroblast (CDKN2A). Those cells can evolve (by doxorubicin in this study) to a condition termed senescence-associated secretory phenotype (SASP) that drives chronic inflammation (termed inflammaging) and fibrosis, determining several chronic lung diseases.  In addition to polyphenol and the above mentioned antifibrotic drug, the hypoglycemic metformin and the mTOR inhibitor rapamycin  are also included in the combinations due to their antiaging potential.

According to that, the study might be interesting and inside the scope of the Journal. Introduction is correct, and references to previous studies related to polyphenols are appropriate.

Methods

SASP was induced by doxorubicin, and this phenotype was determined using the appearance of cell cycle arrest, SA-Gal expression and high IL-6 production as senescence biomarkers. It seems that the choice of these biomarkers is based in data published very recently (2024) by the same research team [ref. 35].  In turn, evaluation of collagen for fibrosis (see lines 365-366)  COL3A, COL1A, and TGF-beta gene expression were also performed as described in reference 35. Additional references of articles published by other groups would be needed to corroborate the validity of most of the methods used. It seems that  all the work is based on methods recently established by the same group in just one paper. The requirement of additional references is also important, as there are no references at all in the results section. Scientifically, this is not the most appropriate.

Results

Just at the beginning of result, line 100: “The main cytokine associated with senescence, IL-6”. It is assumed that the 3 cytokines studies, IL6, IL8 and IL1beta are biomarkers of inflammation, but IL6 as main cytokine for senescence is unclear tome. Some references are needed to give IL6 the most crucial role.

Figure 1 shows that the effects of several polyphenols and other drugs is diverse. Quercetin is the only one increasing IL6, but for instance Gal, Quer and Resv increase a lot the concentration of IL-1beta and possibly inflammation. Figure 2 introduces some combination and honestly the pattern is rather erratic. For instance, caffeic acid was inhibitor of IL1beta, but the combinations of caff with rapamycin is opposite. The above-mentioned effect of Gal, Quer and Resv on IL-1beta disappear after combination of those polyphenols with met of rapamycin. Authors just describe the results, but this is poor.      

Figure 3 is a scheme of the possible effects using combinations of polyphenols and Met/Rapa using green color, but some explanation would be needed. Is that referred only to IL6? Why only IL6? What does dark green or light green mean?  The required improvements for Figure 3 should also be applied to the scheme at Figure 8

Figures 4 and 5: All lines are in blue. It is difficult to see the differences. Please, increase the scale and if possible, introduce new colors to a better observation of the results. They are only referred to caff and metformin. Given the diversity found at Figures 1 and 2, this choice should be justified.

Section 2.2: anti-fibrotic effect. It is headed by 2.2. Anti-fibrotic effects of drugs and polyphenols.

However, only epicatechin showed antifibrotic activity.  This polyphenol is a flavonoid (flavan-3-ol), structurally different of other polyphenols. In subsequent studies, this flavonoid is combined with Met, Rapa and the antifibrotic drugs. The effects are relatively moderate and also diverse. Authors should effort for giving a reasonable analysis of those data, rather than just a description of the diverse effects. Is there any link between the structure of the polyphenol and the anti-inflammatory or antifibrotic effect ?  

Then, section 2.3 is related to EMT evaluation.

In principle, the article is going more and more diverse. The cells used are different, different genes as biomarkers. In this section, only epicatechin as polyphenol is maintained. Other polyphenols are not considered. In my opinion, the article contains too many data, with a lot of diversity, and just descriptions. According to the title, polyphenols would be the more important compounds to be studied, but in fact, they are not. Perhaps at the beginning, but not at the end.

About discussion

In my opinion, this section increases the diversity of data and focusing found that the results. Discussion introduces a lot of genes, proteins and processes that are related to the inflammation or fibrosis, but they are not really determined at the results. For instance, in a few lines, AMPK activation, fatty acid synthesis, inhibition of mTORC1, and inhibition of the production of mitochondrial ROS, inhibition of NFκB transcription factor and so on are mentioned (soon later p16, p21). Only scavenging of mitochondrial ROS, are related to polyphenols. I understand that discussion can introduce some new factors in the cellular metabolism, but in my opinion,  there are too many, and most of them are not directly related to polyphenol and the proposed title of the manuscript. Discussion from line 274 to the end is really focused into the results described earlier. I agree with this part. I recommend to the authors re-think the manuscript to get a more compact and coherent format.  

Finally, I would like to mention that this work found caffeic acid and epicatechin as the most useful polyphenols. However, the scientific literature is full of different data considering quercetin, resveratrol or other as the most effective ones, at least as anti-inflammatory agents. I recognize the difficulty of the issue, but this should be discussed. It is not acceptable that this important issue is just ignored in the current manuscript.

 Minor points

Line 169: is picrosirius red the same than the Sirius red described at M&M?. Please clarify.

Line 184: Replace TGFB1 by TGFb1. Repair the name of this protein at lines 223, 226 and other if necessary.

Line 319: Replace 10 uM by 10 mM,

Line 345: Replace MgCl2 by MgCl2

Author Response

Reviewer 1.

Considering that pirfenidone and nintedanib are antifibrotic drug, and polyphenols are antioxidant and immunomodulatory natural products, this study explores the possible beneficial effect of combinations of those antifibrotic drugs and several polyphenols for preventing cellular senescence of human lung fibroblast (CDKN2A). Those cells can evolve (by doxorubicin in this study) to a condition termed senescence-associated secretory phenotype (SASP) that drives chronic inflammation (termed inflammaging) and fibrosis, determining several chronic lung diseases.

In addition to polyphenol and the above mentioned antifibrotic drug, the hypoglycemic metformin and the mTOR inhibitor rapamycin are also included in the combinations due to their antiaging potential.

According to that, the study might be interesting and inside the scope of the Journal. Introduction is correct, and references to previous studies related to polyphenols are appropriate.

Methods

SASP was induced by doxorubicin, and this phenotype was determined using the appearance of cell cycle arrest, SA-Gal expression and high IL-6 production as senescence biomarkers. It seems that the choice of these biomarkers is based in data published very recently (2024) by the same research team [ref. 35]. In turn, evaluation of collagen for fibrosis (see lines 365-366) COL3A, COL1A, and TGF-beta gene expression were also performed as described in reference 35. Additional references of articles published by other groups would be needed to corroborate the validity of most of the methods used. It seems that all the work is based on methods recently established by the same group in just one paper. The requirement of additional references is also important, as there are no references at all in the results section. Scientifically, this is not the most appropriate.

R: We would like to thank you for your careful reading of the manuscript and criticisms that will contribute to improving its quality of scientific dissemination. Although we used a reference published by our research group, with the data we obtained in the search for combinations of senolytics, there are other references, whether on concepts about cellular senescence or on the methods we used in these two articles (not with the same drugs, but with a similar methodology). Doxorubicin induces DNA damage, promotes SA-β-gal and SASP expression and causes growth arrest in cell lines, and has been used as one of the classic senescence inducers in vitro. We added references in Introduction section and information about doxorubicin model. Note that we standardized the best concentration and analysis times of senescence markers for the MRC-5 lineage under our working conditions in our previous work, thus the reference Godoy et al (2024) best describes the methodology of this work.

Cell cycle arrest analysis, SA-Gal expression and, IL-6 and IL-8 production has been described as classical senescence biomarkers (10.3390/ijms23084168) and because that they are chosen to be evaluated.

We apologize for the error. The cited reference (35) does not describe the fibrosis model methodology or the evaluation of gene expression of fibrosis markers. Reference 35 is important for describing the standardization of the doxorubicin-induced senescence model and the best times for evaluating senescence markers, as well as presenting screening data to determine non-toxic concentrations of polyphenols and some drugs used in this work. These data are in the supplementary material of the article Godoy et al. (2024) - ref 35. It was corrected.

We included the reference for the b-gal protocol (DOI: 10.1007/978-1-62703-239-1_8), Sirius red protocol (DOI: 10.1007/978-1-4939-6430-7_39) and EMT using A549 cells and TGF-b protocol (DOI: 10.1186/s10020-021-00283-6).

In general, we don’t cite references in the Result section, leaving this section only for the description of the results we obtained.

Results

Just at the beginning of result, line 100: “The main cytokine associated with senescence, IL-6”. Itis assumed that the 3 cytokines studies, IL6, IL8 and IL1beta are biomarkers of inflammation, butIL6 as main cytokine for senescence is unclear tome. Some references are needed to give IL6the most crucial role.

R: We added some references in the Discussion section that present IL-6 as an important cytokine of inflammaging.  (doi: 10.3389/fragi.2022.84082; 10.3390/ijms23084168; 10.14283/jpad.2022.42; 10.1016/j.exger.2022.111931)

Figure 1 shows that the effects of several polyphenols and other drugs is diverse. Quercetin isthe only one increasing IL6, but for instance Gal, Quer and Resv increase a lot the concentrationof IL-1beta and possibly inflammation. Figure 2 introduces some combination and honestly the pattern is rather erratic. For instance, caffeic acid was inhibitor of IL1beta, but the combinations of caff with rapamycin is opposite. The above-mentioned effect of Gal, Quer and Resv on IL-1beta disappear after combination of those polyphenols with met of rapamycin. Authors just describe the results, but this is poor.

R: Quercetin not increased IL-6, it was not statistically different from control. Yes, we registered an increase of IL-1b for gal, quer and resv released by senescent cells. When combined caff and rapa, the result is not significant. The amount of IL-1B when MRC-5 was incubated with combinations of gal, quer and resv with metf and rapa was closer to control values but no inhibition was recorded. We agree that IL-1b results are confusing and to some extent, disappointing.

One hypothesis that can be put forward is that the IL-1b release by resveratrol, quercetin and gallic acid observed in the context of cellular senescence of a normal cell (fibroblast) could be associated with apoptosis or autophagy induction, which is described to be induced by polyphenols. We included this hypothesis in the discussion (with references), but more experiments need to be performed to understand this result.

Figure 3 is a scheme of the possible effects using combinations of polyphenols and Met/Rapa using green color, but some explanation would be needed. Is that referred only to IL6? Why onlyIL6? What does dark green or light green mean?

R: The idea of this picture was to show how many SASP cytokines were inhibited by the combinations and isolated compounds, which justified the choice for subsequent experiments. Example: caffeic acid alone inhibited the 3 cytokines evaluated (dark green), metf+caff inhibited 2 cytokines (lighter green), etc. Maybe it was not necessary. We removed the Figures and included this information in the text.

The required improvements for Figure 3 should also be applied to the scheme at Figure 8

R: Both figures were removed.

Figures 4 and 5: All lines are in blue. It is difficult to see the differences. Please, increase the scale and if possible, introduce new colors to a better observation of the results. They are only referred to caff and metformin. Given the diversity found at Figures 1 and 2, this choice should be justified.

R: The choice of caff and metf was explained in the results (line 128-131).

Section 2.2: anti-fibrotic effect.

It is headed by 2.2. Anti-fibrotic effects of drugs and polyphenols.

However, only epicatechin showed antifibrotic activity.

R: We altered the headed title.

This polyphenol is a flavonoid (flavan-3-ol), structurally different of other polyphenols. In subsequent studies, this flavonoid is combined with Met, Rapa and the antifibrotic drugs. The effects are relatively moderate and also diverse. Authors should effort for giving a reasonable analysis of those data, rather than just a description of the diverse effects. Is there any link between the structure of the polyphenol and the anti-inflammatory or antifibrotic effect ?

R: We included considerations in the discussion.

Then, section 2.3 is related to EMT evaluation.

In principle, the article is going more and more diverse. The cells used are different, different genes as biomarkers. In this section, only epicatechin as polyphenol is maintained. Other polyphenols are not considered. In my opinion, the article contains too many data, with a lot of diversity, and just descriptions. According to the title, polyphenols would be the more important compounds to be studied, but in fact, they are not. Perhaps at the beginning, but not at the end.

R:  I hope this new version has clarified why we used two cell lines (one for a senescent fibroblast model and the other for evaluating EMT).

- The markers were chosen according to the experimental protocol under development.

- I believe that polyphenols have been identified as to their importance in the context of senescence and fibrotic processes.

- We only evaluated epicatechin and drugs at the end because it was the only polyphenol to demonstrate antifibrotic effects.

About discussion

In my opinion, this section increases the diversity of data and focusing found that the results. Discussion introduces a lot of genes, proteins and processes that are related to the inflammation or fibrosis, but they are not really determined at the results. For instance, in a few lines, AMPK activation, fatty acid synthesis, inhibition of mTORC1, and inhibition of the production of mitochondrial ROS, inhibition of NFκB transcription factor and so on are mentioned (soon later p16, p21). Only scavenging of mitochondrial ROS, are related to polyphenols. I understand that discussion can introduce some new factors in the cellular metabolism, but in my opinion, there are too many, and most of them are not directly related to polyphenol and the proposed title of the manuscript. Discussion from line 274 to the end is really focused into the results described earlier. I agree with this part. I recommend to the authors re-think the manuscript to get a more compact and coherent format.

Finally, I would like to mention that this work found caffeic acid and epicatechin as the most useful polyphenols. However, the scientific literature is full of different data considering quercetin, resveratrol or other as the most effective ones, at least as anti-inflammatory agents. I recognize the difficulty of the issue, but this should be discussed. It is not acceptable that this important issue is just ignored in the current manuscript.

R: The idea was to rescue what literature reports about the anti-aging activity of metformin and caffeic acid.

It is not uncommon for polyphenols, considering that they are diverse molecules that can be presented in monomeric form or in polymers, to present different biological activities. If all polyphenols were equal and had the same biological effects, any combination of dasatinib with a polyphenol would have senolytic action, and this is not true, only quercetin showed potential for this. Our previous work has corroborated it (Godoy et al, 2024). Despite we even identified two other senolytic candidates, the activity was not recorded for all compounds tested. Perhaps resveratrol is more studied, more consumed, but this does not guarantee that it has all the bioactivities researched.

Minor points

Line 169: is picrosirius red the same than the Sirius red described at M&M?. Please clarify.

R: It was correct.

Line 184: Replace TGFB1 by TGFb1. Repair the name of this protein at lines 223, 226 and other if necessary.

R: It was carefully revised. We used TGF-b for protein and TGFB1 for human gene denomination.

Line 345: Replace MgCl2 by MgCl2.

R: It was altered.

Reviewer 2 Report

Comments and Suggestions for Authors

·         The type of study design (in vitro) should be clearly stated in abstract.

·         Line 47-48: Need to mention that, as a result these diseases could promote early-stage lung cancer.

·     Please clearly define/describe cell line A549 somewhere in the introduction. For example, adenocarcinoma  A549 cells are regarded as a model of malignant alveolar type II epithelial cells, which enriched in proteins related to cellular.....

·         Line 82-90: I would suggest expanding this paragraph with the inclusion of in vivo and in vitro studies on the effects of quercetin and other compounds in improving cardiac fibrosis.

·         Line 91-97: The research gap and the associated novelty/contribution aspect of the paper are very weak. It is unclear what this paper adds that is not already known. Why these polyphenolic compounds and drugs, in particular? Why it is important to conduct the experiment in cell line A549? What about other types (e.g., H460).

·         Line 321-322: It would be benefit to include references to clarify toxic doses of drugs in different in vitro cell cultures, particularly A549 cells.

·         All figures should be enlarged to be more clear to the reader.

·         Please define all abbreviations in the first use (e.g., TGF-β).

·         It does not appear that the authors noted any limitations of the study. This should be in a separate section.

·         The conclusions section is missing. It would benefit from giving consideration to how the paragraphs are structured and the thesis of each paragraph. It would also be helpful to include the implications of the findings and their potential impact on future studies.

·         Please include a list of abbreviations at the end.

Comments on the Quality of English Language

Minor edit required.

Author Response

Reviewer 2

The type of study design (in vitro) should be clearly stated in abstract.

R: We included it in Abstract.

Line 47-48: Need to mention that, as a result these diseases could promote early-stage lung cancer.

R: We included a mention to relationship between SASP and lung cancer induction, and a reference.

Please clearly define/describe cell line A549 somewhere in the introduction. For example, adenocarcinoma A549 cells are regarded as a model of malignant alveolar type II epithelial cells, which enriched in proteins related to cellular.....

R: We have included a final paragraph in the Introduction section explaining the models employed. We included a reference to the method that indicates the use of the A549 cell line in EMT assays and included in the description of the cell line that these are human lung carcinoma cells.

Line 82-90: I would suggest expanding this paragraph with the inclusion of in vivo and in vitro studies on the effects of quercetin and other compounds in improving cardiac fibrosis.

R: The paragraph was expanded with a better description of literature data.

Line 91-97: The research gap and the associated novelty/contribution aspect of the paper are very weak. It is unclear what this paper adds that is not already known. Why these polyphenolic compounds and drugs, in particular? Why it is important to conduct the experiment in cell line A549? What about other types (e.g., H460).

R: We agree that it was not clear why the brief trial with the A549 lineage was carried out. We hope this has been clarified now.

The idea was to bring together already known drugs and polyphenols seeking synergistic actions that could increase their potential for therapeutic use, as demonstrated by the combination of dasatinib and quercetin, the combination of senolytics in clinical studies. In this reasoning, we start from pharmacological activities already described for drugs (anti-aging and anti-fibrotics) associating them with different polyphenols. We innovated mainly by revealing the activity of caffeic acid as a senostatic.

Line 321-322: It would be benefit to include references to clarify toxic doses of drugs in different in vitro cell cultures, particularly A549 cells.

R: We included our data about cytotoxicity absence in cell lines used as Supplementary material.

All figures should be enlarged to be more clear to the reader.

R: It was enlarged.

Please define all abbreviations in the first use (e.g., TGF-β).

R: We read carefully and included terms in full in the text. We have also compiled a list of abbreviations at the end of the manuscript.

It does not appear that the authors noted any limitations of the study. This should be in a separate section.

R: We included some considerations about the limitations of the study in the Discussion section.

The conclusions section is missing. It would benefit from giving consideration to how the paragraphs are structured and the thesis of each paragraph. It would also be helpful to include the implications of the findings and their potential impact on future studies.

R: The conclusions are provided as the last paragraph of the Discussion section.

Please include a list of abbreviations at the end.

R: It was included.

Round 2

Reviewer 1 Report

Comments and Suggestions for Authors

Authors have submitted a reply letter addressing all the points and suggestions contained in the previous report, and the manuscript has been modified according to those points. A number of references have been added (from 35 to 49 in the amended version) and some sections of the manuscript are more documented. Both the description of the methods and rationality of the work concerning the convenience of the biomarkers of senescence-associated secretory phenotype (SASP) studied have been improved. The combination of polyphenols with metformin and antifibrotic drugs is justified, as well as the selection of the most effective polyphenols as long as the studies proceeded. I agree that the relative efficiency of polyphenols is a complex issue, and different reports found different order of antioxidant and anti-inflammatory activity. Figures have been simplified. Minor errors and typos have been corrected. The follow up of the work is more consistent and the link between inflammation and fibrosis is more logical, so that the diversity of the biomolecules studied (polyphenols, antiaging and antifibrotic drugs) has gained comprehension.   

Reviewer 2 Report

Comments and Suggestions for Authors

No further comments.